# Nafcillin-Loaded Photocrosslinkable Nanocomposite Hydrogels for Biomedical Applications

**DOI:** 10.3390/pharmaceutics15061588

**Published:** 2023-05-24

**Authors:** Gabriela Toader, Ionela Alice Podaru, Edina Rusen, Aurel Diacon, Raluca Elena Ginghina, Mioara Alexandru, Florina Lucica Zorila, Ana Mihaela Gavrila, Bogdan Trica, Traian Rotariu, Mariana Ionita

**Affiliations:** 1Military Technical Academy, “Ferdinand I”, 39-49 G. Cosbuc Blvd., 050141 Bucharest, Romania; nitagabriela.t@gmail.com (G.T.); podaru.alice04@gmail.com (I.A.P.); aurel_diacon@yahoo.com (A.D.); trotariu@mta.ro (T.R.); 2Faculty of Chemical Engineering and Biotechnologies, University Politehnica of Bucharest, 1–7 Gh. Polizu Street, 011061 Bucharest, Romania; mariana.ionita@polimi.it; 3Research and Innovation Centre for CBRN Defense and Ecology, 225 Şos. Olteniţei, 041327 Bucharest, Romania; raluca.ginghina@nbce.ro; 4Microbiology Laboratory, Horia Hulubei National Institute for R&D in Physics and Nuclear Engineering, 30 Reactorului St., 077125 Magurele, Romania; florina.zorila@nipne.ro; 5Department of Genetics, Faculty of Biology, University of Bucharest, 91-95 Splaiul Indepententei, 050095 Bucharest, Romania; 6National Institute of Research, Development for Chemistry and Petrochemistry, 202 Splaiul Independentei, 060041 Bucharest, Romania; ana.gavrila@icechim.ro (A.M.G.); bogdan.trica@icechim.ro (B.T.); 7eBio-Hub Research Centre, University Politehnica of Bucharest-Campus, Iuliu Maniu 6, 061344 Bucharest, Romania

**Keywords:** photocrosslinkable, hydrogel, nafcillin, drug delivery, nanocomposite

## Abstract

Skin infections are frequently treated via intravenous or oral administration of antibiotics, which can lead to serious adverse effects and may sometimes contribute to the proliferation of resistant bacterial strains. Skin represents a convenient pathway for delivering therapeutic compounds, ensured by the high number of blood vessels and amount of lymphatic fluids in the cutaneous tissues, which are systematically connected to the rest of the body. This study provides a novel, straightforward method to obtain nafcillin-loaded photocrosslinkable nanocomposite hydrogels and demonstrates their performance as drug carriers and antimicrobial efficacy against Gram-positive bacteria. The novel formulations obtained, based on polyvinylpyrrolidone, tri(ethylene glycol) divinyl ether crosslinker, hydrophilic bentonite nanoclay, and/or two types of photoactive (TiO_2_ and ZnO) nanofillers, were characterized using various analytical methods (transmission electron microscopy (TEM), scanning electron microscopy–energy-dispersive X-ray analysis (SEM-EDX), mechanical tests (tension, compression, and shear), ultraviolet-visible spectroscopy (UV-Vis), swelling investigations, and via specific microbiological assays (“agar disc diffusion method” and “time-kill test”). The results reveal that the nanocomposite hydrogel possessed high mechanical resistance, good swelling abilities, and good antimicrobial activity, demonstrating a decrease in the bacteria growth between 3log_10_ and 2log_10_ after one hour of direct contact with *S. aureus.*

## 1. Introduction

A wound is a pathological condition produced by injury, burn, infection, or physiochemical alterations of the cutaneous tissue [1]. Wound healing is a complicated regenerative process that involves many distinct tissues and cell lineages [2,3]. Bandages and patches are the first aid for injured tissues in hospitals, on the battlefield, or at home [4,5] for stopping bleeding and contamination with dirt and bacteria [4,5]. Applied on the wound, they have direct contact with the soft wounded tissues, where exudate and nutrients from the blood can nurture the multiplication of bacteria and formation of biofilm [6,7]. If the wounds are not timely dressed, the latter processes usually end up as serious infections. Various solutions, gels, creams, and bandages are used in hospitals to treat wounds [4]. The development of modern materials and new manufacturing technologies have contributed to a new generation of dressings that offer proper conditions for wound healing. Therefore, using an artificial matrix as a skin dressing is nowadays a potential therapeutic option for promoting wound healing. Many dermal dressings and skin substitutes have been created to mimic the wound-healing microenvironment [8]. In addition, there are numerous commercially available hydrogel dressings for wound treatment, ranging from amorphous hydrogels for cavity wounds to hybrid structures such as gel-impregnated plasters for superficial wound care [9]. The materials designed as dressings for wound healing should meet a series of criteria: must be biocompatible, should constitute an efficient barrier for microorganisms [10], possess high antimicrobial activity [11], ensure an adequate drug release pattern [12], have proper absorptive capacity to remove excess exudate from the wound surface while keeping adequate hydration of the tissue [13,14], and have good mechanical resistance and low adherence to the wound bed [15] to prevent additional damage to sensitive skin [14]. In many cases, homopolymers cannot satisfy these diverse demands regarding the abovementioned characteristics and performances [13]; therefore, it is desirable to utilize composite hydrogels or interpenetrated polymeric networks (IPNs) for obtaining performant dressings able to promote the wound healing process [1,13].

Hydrogels are considered a standard therapeutic approach for sloughy or necrotic wounds, because by rehydrating nonviable tissue, hydrogels aid in natural autolysis, facilitate wound debridement, and prevent bacterial infiltration [16]. Yet, they should not be used on wounds that exude an excessive amount of fluid or gangrenous tissue (which should be kept dry to minimize the possibility of infection) [15]. Hydrogels provide the optimal moistness conditions for healing while protecting the wound, with the added benefit of being comfortable for the patient because of their cooling action and nonadhesiveness to damaged tissue [14]. Commonly used polymers in hydrogels for wound healing are gelatin, collagen, chitosan, dextran, alginate, cellulose, polyethylene glycol, polyvinylalcohol, polyvinylpyrrolidone, and acrylic polymers (Carbopol^®^) [9,17,18,19,20]. Polyvinylpyrrolidone (PVP) is a hydrophilic polymer frequently used as a carrier in the pharmaceutical and biomedical fields [21]. PVP is a polymer with enormous potential for producing medicinal formulations due to the fact of its versatility and unique attributes. PVP is also nontoxic and biocompatible; hence, it is suitable for biomedical applications [22]. It has previously been successfully employed in developing several drug delivery systems, including oral, topical, transdermal, and ophthalmic administration [21]. PVP has already been successfully used as a starting material in manufacturing hydrogels for wound dressing [23]. The physical and mechanical properties of PVP-based hydrogels can be adjusted by varying the pH, ionic strength, and gelation temperature [9]. Although they can maintain a moisture balance at the wound site, most hydrogels have poor mechanical properties, limiting their biomedical applications [12]. Composite hydrogels have shown better potential for skin patches because of their straightforward design, ease of preparation, and specific features [8]. Composite hydrogels have attracted interest as wound adjuvants due to the fact of their high mechanical resistance, high porosity, interconnected macroporous network, and large specific surface area, as well as their ability to maintain a humid microenvironment and absorb tissue exudates [8].

Photocrosslinkable hydrogels, in particular, have been extensively researched in biomedical domains, particularly for 3D tissue-engineered structures [14], biosensing mediums [24,25], and drug-controlled release matrices [26]. Compared with other crosslinking techniques, the major advantage of free radical photopolymerization is that hydrogels with a more stable 3D structure and higher stiffness and strength can be produced with the chemical crosslinking resulting from the photoinitiated unsaturated double-bond polymerization [27]. Photoinitiated radical polymerization of multifunctional monomers in the presence of photoactive nanoparticles offers a fast and convenient method for producing highly crosslinked reinforced polymeric networks [28] that can be further employed for wound healing applications. The bacterial response to the activity of the nanoparticles (NPs) is influenced by a series of factors including the number, shape, charge and size of the NPs, and their mechanism of action. The inherent antioxidant defense system of bacteria, which interacts with stressors, is also essential; still, bacterial cells find it highly challenging to develop resistance to NPs, because they simultaneously act against them through distinct efficient mechanisms [4]. It has been found that NPs like TiO_2_ or ZnO can affect bacterial cells by altering the cellular membrane [29]. Zhang M. et al. [30] reported that ZnO NPs enhanced the antibacterial effect of the composite hydrogel aimed at accelerating wound healing. Palantöken A. et al. [31] developed TiO_2_-loaded hydrogels with long-lasting antibacterial activity toward Gram-negative bacteria. Silver nanoparticles are widely recommended for their broad-spectrum antibacterial activity, but their instability may sometimes diminish their efficacy [32]. Nevertheless, often nanoparticles alone are insufficient to reduce the bacterial population at the wound site. The combination of antibiotics with NPs has been revealed to be a more promising approach and an effective tool against bacterial infection. Wang et al. reported a hydrogel with enhanced antibacterial activity obtained by the co-delivery of vancomycin and silver nanoparticles [8,33].

When skin injuries occur, microorganisms can easily invade and cause severe wound infections, thus preventing wound healing [8]; therefore, the presence of powerful antimicrobial ingredients in the composition of the wound dressings is mandatory. The literature reports promising results on the controlled release of antibiotics from hydrogels or encapsulated hybrid nanomaterials [16,34]. Controlled release systems can deliver drugs over an extended period of time at a relatively constant release rate [13]. Antibiotic-loaded hydrogels are an innovative approach for improving medical treatment, decreasing overall healthcare costs, increasing the therapeutic effect of drugs, and balancing the toxicity of medications commonly employed for wound care [35]. Furthermore, they are crucial in treating local infections when high antibiotic concentrations are required [36]. Ideally, the diffusion paths of the drug molecule; thus, the release rate of the active ingredients can be controlled by modifying the characteristics of the polymeric network [37]. Antibiotics loaded into transdermal dressings for local wound dressing protocols may provide therapeutic antibacterial effects while being absorbed into the body [38]. Furthermore, local delivery can reduce the possibility of systemic antibiotic absorption, reducing antibiotic resistance [38]. On the other hand, the drug concentration in the hydrogel should be strictly controlled, since high amounts of antibiotics can lead to systemic toxicity [39]. Some antibiotic-loaded hydrogels demonstrated they could help treat chronic infectious lesions during wound closures due to the fact of their excellent antimicrobial activity against Gram-positive or Gram-negative strains [40]. Beta-lactam antibiotics (amoxicillin and ampicillin) and aminoglycosides (kanamycin) are well-known drugs applied to inhibit bacterial spread in wound-dressing applications [38]. Katime I. et al. [41] described nafcillin-loaded copolymeric poly(acrylic acid-co-methyl methacrylate) hydrogels. Si H. et al. [42] reported a 3D printed hyaluronic-acid-based double-crosslinked hydrogel with incorporated nafcillin. Nafcillin is a semisynthetic antibiotic considered a narrow-spectrum beta-lactam antibiotic [43]. Nafcillin is a beta-lactamase-resistant penicillin indicated for treating staphylococcal infections caused by strains resistant to other penicillins [44]. It may also be used as a first-line therapy in methicillin-sensitive *Staphylococcus aureus* infections [44]. Studies comparing clinical treatment results with vancomycin suggest that the greater efficacy of nafcillin against methicillin-susceptible *S. aureus* (MSSA) may be related to its capacity to enhance sensitivity to innate host defense peptides (HDPs) [45]. Composite hydrogel-based dressings are recommended for treating damaged cutaneous tissues because they facilitate and accelerate wound healing due to the fact their unique attributes. However, these materials alone do not possess antimicrobial activity, so adding active ingredients is required to achieve antimicrobial efficacy. The most common solution is to load antimicrobial agents, antibiotics, and/or active nanoparticles into the hydrogel matrix to treat the wound site efficiently. Various carrier matrices and drug loading methods have been developed over the last decades. Interfacial polymerization, solvent casting, phase inversion, or emulsion polymerization are the most often utilized polymerization techniques to obtain polymer matrixes that could serve as drug carriers [46]. Even though numerous transdermal drug-delivering systems have been widely reported, some of their drawbacks are related to the laborious manufacturing processes, costs, limitations in chemical resistance and processability, or difficulty in maintaining a constant product quality throughout the whole manufacturing process [46]. These disadvantages could be remediated by developing photocrosslinkable polymer solutions, which ensure a faster, replicable manufacturing process and ease of processability.

In this context, this study presents an innovative, straightforward method of obtaining nafcillin-loaded photocrosslinkable nanocomposite hydrogels potentially suitable for cutaneous application. Novel formulations based on photocrosslinked PVP chains, interconnected by tri(ethylene glycol) divinyl ether (TEGDVE) crosslinker and reinforced by hydrophilic bentonite nanoclay and/or two types of photoactive nanoparticles, TiO_2_ and ZnO, were loaded with nafcillin. Further, we sought to explore the influence of each formulation on the properties of the resultant nanocomposite hydrogel, evaluate the nafcillin release profile, and demonstrate their antimicrobial effect via in vitro microbiological assays. The novelty of this study resides in the ease of fabrication of the antibiotic-loaded nanocomposite films, as well as the benefits provided by their biocompatible components and efficacy against methicillin-susceptible *Staphylococcus aureus* due to the systematic release of nafcillin.

The herein reported nanocomposite hydrogels demonstrated that they absorb fluid efficiently, are comfortable to touch and easy to remove, have high elasticity but also remarkable mechanical strength, and can operate as a barrier against pathogens, and possessing good antimicrobial activity against Gram-positive strains as a result of the efficient nafcillin-targeted release.

## 2. Materials and Methods

### 2.1. Materials

Materials used for the synthesis of the photocrosslinkable nanocomposite hydrogels: The monomer 1-vinyl-2-pyrrolidone (NVP) (sodium hydroxide as an inhibitor, ≥99%, Sigma Aldrich, St. Louis, MO, USA) was distilled before use with vacuum distillation (90–92 °C at 10 mmHg) and stored under N_2_ before being employed to synthesize the hydrogel. Tri(ethylene glycol) divinyl ether (TEGDVE) (Sigma Aldrich, St. Louis, MO, USA), 2-hydroxy-4′-(2-hydroxyethoxy)-2-methylpropiophenone (Ph-In, Sigma Aldrich, St. Louis, MO, USA), and bentonite (BT) (hydrophilic bentonite nanoclay, Nanomer^®^ PGV, Sigma Aldrich, St. Louis, MO, USA) were used as received. The nanofillers, TiO_2_ and ZnO, were synthesized as elsewhere described [47]. Materials used for drug loading/release: nafcillin sodium salt monohydrate (nafcillin) (Sigma-Aldrich, St. Louis, MO, USA) and phosphate-buffered saline (PBS) (Sigma-Aldrich, St. Louis, MO, USA). Materials used for evaluation of the antimicrobial activity: chloramphenicol (Sigma-Aldrich, St. Louis, MO, USA), Muller Hinton broth (MHb) (Merck, Darmstadt, Germany), Mueller Hinton agar (Merck), and 90 mm diameter Petri dishes. The bacteria strains were *Staphylococcus aureus* (ATCC 6538) as a model for Gram-positive bacteria and *Escherichia coli* (ATCC 8739) as a model for Gram-negative bacteria. *S. aureus* and *E. coli* were chosen, as they are considered standard microorganisms for testing the antimicrobial properties of newly synthesized products [48]. After cultivation overnight in MHb (Merck) at 37 °C with stirring (200 rpm), the bacterial strains were harvested. Portions of the suspension were harvested by centrifugation and resuspended in phosphate buffer saline. The suspensions were adjusted to approximately 10^7^ CFU/mL [49].

### 2.2. Methods

#### 2.2.1. Synthesis of Photocrosslinkable Nanocomposite Hydrogels

The nanocomposite hydrogels were obtained with free radical photopolymerization in silicone rubber-sealed glass molds using a UV lamp (low-pressure Hg UV lamp, λ_em_ = 254 nm). The nanofiller composition and the sample codes for the synthesized materials are detailed in Table 1. For the P1 sample, the monomer NVP (1.2 g), the crosslinker–TEGDVE (0.09 g), and the photoinitiator-Ph-In (0.006 g) were dissolved in distilled water (4.5 mL). To obtain the nanocomposite hydrogels, the nanofillers, BT and TiO_2_/ZnO (compositions detailed in Table 1), were firstly dispersed through ultrasonication, followed by the addition of the other reactants. The UV curing of the hydrogels was completed after approximately 30 min.

#### 2.2.2. Nafcillin Loading/Release Tests

The lyophilized nanocomposite hydrogels were loaded with nafcillin to investigate the release profiles of the active ingredient for each formulation. An aqueous nafcillin solution (5 × 10^−3^ M) was utilized for loading. Each lyophilized sample (~30 mg) was introduced in 5 mL of nafcillin solution and maintained in this solution pending equilibrium. The nafcillin-loaded equilibrium-swollen nanocomposite hydrogels were dried in an oven to reach a constant weight. The nafcillin-loaded dry samples were further subjected to release tests in PBS solution (pH = 7.4, 37 °C). The UV-Vis technique was employed for observing the nafcillin release profile in PBS. The absorbance at 330 nm was monitored by recurrent sampling (1 mL) at regular intervals and, subsequently, correlated with the corresponding concentration of nafcillin. The solution in the test tube was refilled with 1 mL of fresh solvent after sampling to maintain a constant volume for the release experiments. The data obtained from the UV-Vis analysis were used to evaluate the release patterns of the nanocomposite hydrogels by investigating which mathematical model from the ones listed below was more appropriate for describing the drug release profile. The accuracy of the fit was also estimated. Measurements were performed in duplicate, and the mean values were reported. Thus, the drug release profile was fitted using the four mathematical models presented bellow [16].

Zero order (Equation (1)):(1)$$Q\%=K_{0}t$$

First order (Equation (2)):(2)$$Q\%=1-e^{K_{1}t}$$

Simplified Higuchi model (Equation (3)):(3)$$Q\%=K_{H}t^{0.5}$$

Linear logarithm form of the Korsmeyer–Peppas model (Equation (4)):(4)log(Q%)=log(K)+nlog(t)
where *Q*% is the percentage of drug released in time t; *K*_0_ is the zero-order release constant in units of concentration %/time; *K*_1_ is the first-order release constant; *K*_H_ is the Higuchi dissolution constant; *K* is the kinetic constant characteristic of the drug–polymer system; and *n* is the release coefficient that indicates the type of diffusion mechanism.

#### 2.2.3. Evaluation of the Antimicrobial Activity of the Nafcillin-Loaded Nanocomposite Hydrogels

Nafcillin, like other penicillins, has bactericidal activity against penicillin-susceptible microorganisms during the state of active multiplication in the bacterial cell wall synthesis. It acts by inhibition of the biosynthesis of the bacterial cell wall by forming covalent bonds with penicillin-binding proteins that play a critical role in the final transpeptidation process. It inhibits transpeptidase and carboxypeptidase activities by binding to penicillin-binding proteins conferred by these proteins and prevents the formation of the crosslinks [50].

The antimicrobial activity of the nafcillin-loaded nanocomposite hydrogels was evaluated with the “agar disc diffusion method” and the “time-kill test”. The samples were utilized for in vitro evaluation against *Staphylococcus aureus* and *Escherichia coli* bacteria.

The disc diffusion method is the official method used in many clinical microbiology laboratories for routine antimicrobial susceptibility testing. In this well-known procedure, agar plates are inoculated with a standardized inoculum (10^5^ CFU/mL) of the test microorganism. Then, filter paper discs (~6 mm diameter) containing the test compound at a known concentration are placed on the agar surface. The Petri dishes are incubated under suitable conditions (36 °C for 24 h). Yet, the agar disc diffusion method is not appropriate to determine the minimum inhibitory concentration (MIC), as it is impossible to quantify the amount of the antimicrobial agent diffused into the agar medium [49]. In this study, due to the nature and desired applicability of the hydrogels (nafcillin-loaded solid samples designed for cutaneous healing applications), the nanocomposite hydrogel films (~0.3 × 0.5 × 0.3 cm) were applied directly onto the agar inoculated with *S. aureus* or *E. coli*. If the tested materials have antimicrobial activity, an inhibition area should be observed around the sample on the agar surface. The antimicrobial efficiency of the samples was compared with the positive control consisting of a standard disc with chloramphenicol.

Time-kill assay is a method employed for quantifying the bactericidal effect. This method is suitable for obtaining information about the antimicrobial agent and microbial strain interaction. The time-kill test reveals a time-dependent or a concentration-dependent antimicrobial effect. Portions of 0.2 × 0.2 cm from each sample were put in contact with 1 mL of inoculum from each microorganism (bacterial strain suspensions, 10^7^ CFU/mL). The samples were mixed and vortexed with inoculum until obtaining a homogenous mixture. They were maintained in direct contact (for 1 h and 24 h). At each established time, a volume of 0.1 mL of each mixture was inoculated onto the agar surface and incubated at 37 °C for 24 h. After the incubation, bacterial survival was evaluated [51].

### 2.3. Characterization

The FTIR analysis was performed on a Spectrum Two FTIR spectrometer (PerkinElmer, Waltham, MA, USA) with a MIRacleTM Single Reflection ATR-PIKE Technologies at 4 cm^−1^ resolution, summing 16 scans in the 4000−550 cm^−1^ region. The TEM imaging of the nanofillers was obtained with a TECNAI F30 G2STWIN instrument, Fei Company, Oregon, OR, USA, at a 300 kV acceleration voltage and 1 Å resolution. The cross-sections of the equilibrium-swollen hydrogels were subjected to a freeze–drying cycle at −15 °C using Biobase BK-FD10 S equipment. Before the SEM-EDX analysis, the lyophilized samples were gold sputter-coated under argon plasma. SEM imaging was acquired using a Tescan Vega II LMU electronic microscope at a 30 keV acceleration voltage, and the elemental mapping was assessed with the EDX technique (Bruker QuantaxXFlash 6/10 energy-dispersive X-ray). Operational parameters: voltage, 30 keV; optimum scan current, 20 ÷ 40 nanoamperes; resolution, 10 nm; pressure, 50 × 10^−2^ Pa. The swelling degree of the nanocomposite hydrogels was estimated according to References [52,53,54], based on duplicate measurements, and the mean values are reported.

The swelling degree (SD) was calculated according to the equation described below:*SD* = (*w_hydrogel_* – *w_xerogel_*)/*w_xerogel_*
where *w_hydrogel_* is the weight of the hydrogel (at equilibrium swollen state), and *w_xerogel_* represents the weight of the xerogel (completely dried matrix).

A Discovery 850 DMA TA analyzer was utilized to investigate the mechanical properties of the nanocomposite hydrogels through uniaxial tensile/compressive testing and the frequency sweep on the “shear-sandwich” geometry. The tensile tests were performed to investigate the mechanical resistance of the synthesized nanocomposite hydrogels to tensile deformation at a rate of 5 mm/min. After the UV-Vis curing procedure was completed, five specimens (40 × 5 × 3 mm) from each sample were evaluated for tensile resistance using tension clamps, and mean values were recorded. Compressive tests (compression clamps: Ø 40 mm) were performed at 2 mm/min on five fully swollen disc specimens from each type of sample, and the mean values are reported. “Shear-sandwich” clamps were installed on the same DMA 850 TA instrument to evaluate the viscoelastic properties of the nanocomposite hydrogels in the frequency-sweep mode, in the linear visco-elastic region, maintaining a constant strain of 1%. At the same time, the frequency range was logarithmically increased from 0.1 to 10 Hz. For the frequency sweep survey, two equilibrium-swollen square-shaped specimens (10 × 10 × 3 mm) were tested on the “shear-sandwich” set-up, and the mean values are reported. The UV-Vis survey was performed in the 300–800 nm range, with a resolution of 5 nm, using a GBC Cintra 101 UV-Vis instrument.

## 3. Results and Discussions

### 3.1. Preparation of the Nanocomposite Hydrogels

The nanocomposite hydrogels were obtained via free-radical photopolymerization of NVP by employing TEGDVE as the crosslinking agent, and BT and TiO_2_/ZnO as nanofillers. Figure 1 displays a schematic representation of the main steps followed to obtain the nafcillin-loaded nanocomposite hydrogels and a brief exemplification of one of their potential applications. These steps are further detailed, starting with a description of the materials and the procedures utilized for synthesizing and characterizing the nanocomposite hydrogels, followed by nafcillin loading, and then investigations of the drug release kinetics and microbiological assay.

The selection of each reactant used for synthesizing the nanocomposite hydrogels is further explained. Due to the versatile and unique properties of PNVP adjoined to the countless possibilities for use in drug delivery systems [21], NVP was the favorite monomer candidate for this study. TEGDVE is frequently used as a crosslinking agent due to the fact of its high reactivity, compatibility with UV curing devices, and molecule flexibility [55,56,57,58,59]. TiO_2_ and ZnO nanofillers were chosen for their antimicrobial potential and photoactivity [60,61]. Hydrophilic BT nanoclay was employed as a reinforcing agent in the hydrogels due to the fact of its remarkable ability to enhance mechanical properties [55,62]. Still, it is also well known for its cationic exchange capacity, swelling and drug-carrier abilities, and possible applications for improving antimicrobial activity [63].

UV-assisted synthesis of the nanocomposite hydrogels represents the first stage of this research, and the critical parameter, in this case, was achieving a homogenous dispersion of the nanofillers. Therefore, before adding the monomer to the reaction mixture, the formulations were subjected to an hour-long ultrasound dispersion process to achieve a uniform distribution of the nanoparticles. After UV curing, the nanocomposite hydrogel films were removed from the glass molds and purified in distilled water. Since these materials were intended for biomedical applications, this purification step was essential to remove the unreacted components. Finally, the purified equilibrium-swollen samples were further submitted to a freeze–drying process to yield porous structures that were subsequently loaded with nafcillin, a penicillin derivative antibiotic usually employed to treat staphylococcal infections.

### 3.2. Characterization of the Nanocomposite Hydrogels

The FTIR analysis was used to evaluate the obtained xerogels (Appendix A). The analysis confirmed the absence of monomers (the specific vibration of C=C observed for NVP and TEGDVE at 1620 cm^−1^ was missing) in the xerogels and that the spectra displayed the characteristic signals for CH (2920 cm^−1^), C=O (1670 cm^−1^), C-N (1420 cm^−1^), and C-O (1280 cm^−1^) specific for the crosslinked polymer chains including NVP units. The presence of TiO_2_ and ZnO in the composites could not be properly ascertained by FTIR, probably due to the fact of their wrapping in the polymer matrix.

Electron microscopy techniques, TEM and SEM (with EDX), were performed on the nanocomposite hydrogels to ascertain the morphology and distribution of the filler in the polymer matrix.

The TEM images in Figure 2 offer evidence of the geometric shape and the dimensions of the TiO_2_ and ZnO nanoparticles utilized in the nanocomposite hydrogels. As can be seen, TiO_2_ displayed a relatively round, regular shape, while the ZnO nanoparticles possessed an irregular shape and relatively smaller dimensions than TiO_2_. Considering that each larger particle seemed to be composed of a few smaller crystallites, we can affirm that the TEM pictures also suggest that some of the TiO_2_ and ZnO nanoparticles were aggregated and assembled within similarly shaped larger particles. The Influence of the morphology of the TiO_2_ and ZnO crystals on the performances of the resulting nanocomposite hydrogels is further discussed through each supplementary investigation performed.

The morphology of the lyophilized and synthesized nanocomposites was revealed with SEM-EDX analysis (Figure 3 and Appendix A).

Since the amount of monomer and crosslinker employed for the synthesis was kept constant, modification of the hydrogel pore characteristics can be attributed to the modifications induced by the type and combination of the nanofiller. Comparing the SEM images of samples P1 and P2, the increase in the pore size for sample P2 can be attributed to an increase in the morphology’s anisotropy caused by obtaining exfoliated or intercalated morphologies induced by the presence of bentonite (Figure 1).

In the case of samples P1-TiO_2_ (Figure 3C) and P1-ZnO (Figure 3E), there was no apparent difference between the pores; however, slightly larger dimensions were obtained compared to the blank sample P1. This behavior can be attributed to the participation of the TiO_2_ and ZnO nanoparticles in the photopolymerization step through the generation of HO free radical species induced by UV irradiation [64,65]. The increase in free radical concentration leads to an increase in the polymerization rate, and an increase in the volume contraction, which causes the formation of larger pores during the relaxation step [66]. Furthermore, the pore size dimension in the case of P2-TiO_2_ and P2-ZnO was 15–20 times larger than the sample without photoactive particles. This can be attributed to the presence of bentonite, which undergoes a more efficient exfoliation process (supported by the presence of TiO_2_ or ZnO, Figure 1) compared to the P2 sample. The EDX mapping (Appendix A) revealed the uniform distribution of the nanofillers inside the nanocomposite hydrogels P1-TiO_2_, P1-ZnO, P2-TiO_2_, and P2-ZnO, respectively.

The following parameter investigated and correlated with the SEM-EDX analysis was the swelling degree of the nanocomposite hydrogels. In Figure 4, the variation in the swelling degree is presented. Sample P1 exhibited the highest swelling degree. The water retention capacity decreased with the increase in nanofillers, indicating a morphological modification through the formation of agglomerates in the polymer network. Comparing the samples with ZnO and TiO_2_ fillers, the higher swelling degree of the sample containing TiO_2_ was evident, which can be attributed to its higher hydrophilicity compared to the sample containing ZnO. In the case of sample P2, the slightly opaquer aspect compared to P1 is explained by the presence of bentonite nanoclay. Comparing P1-TiO_2_ and P1-ZnO, in the case of ZnO, the dispersion of the nanoparticles seemed more efficient, which can be attributed to the smaller dimensions of the nanoparticles sustained by the EDX mapping (Appendix A). In the case of P2-TiO_2_ and P2-ZnO, the loss of transparency (Figure 4B) was due to an increased number of reinforcing agents.

Tensile/compressive and frequency sweep measurements were used to investigate the mechanical properties of the nanocomposite hydrogels. Figure 5 comparatively illustrates the results obtained from these investigations.

The tensile tests (Figure 5A), performed after the completion of the UV curing, showed that the addition of the nanofillers considerably improved the mechanical resistance of the blank sample (P1). The mechanical properties of samples P1-TiO_2_ and P1-ZnO were superior to P1, attributable to the use of reinforcing agents. It is well documented in the literature that the addition of bentonite has a positive effect on mechanical properties, an aspect clearly visible for the analyzed samples. The positive impact of BT on the mechanical properties [67,68] of sample P2 was also sustained by the SEM images, which revealed that sample P2 possessed thicker pore walls than the blank sample (P1); thus, a reinforcing effect was confirmed by an increase in the mechanical stiffness [69] of sample P2. The mean ultimate strain values measured for P2 were slightly lower, probably because this stiffening effect was induced by the nanoclay, meaning that it was harder to deform the hydrogel [70]. In the case of P2-ZnO, the mechanical properties were inferior to P2-TiO_2_, explainable by the interaction between the nanoparticle and the polymer network (hydrophilic/hydrophobic interactions) (Figure 1).

The results obtained from the uniaxial compressive tests performed on the equilibrium-swollen nanocomposite hydrogels are comparatively illustrated in Figure 5B. The swollen state of the crosslinked samples led to the slightly different behavior of the hydrogels due to the fact of their distinct water uptake capacity in addition to the influence caused by the mechanical resistance of each type of nanofiller. Still, there is a delicate balance in designing a hydrogel with appropriate properties that exhibit sufficient mechanical integrity without sacrificing its absorptive properties [71].

Compressive testing showed that the P1-ZnO sample, in a fully swollen state, lost its resistance more than the analogous sample with TiO_2_ (P1-TiO_2_). A possible explanation for this behavior could be the better dispersion of TiO_2_, probably sustained by its higher hydrophilicity compared to ZnO, conjoined with the potential, more pronounced, aggregation of the smaller ZnO nanoparticles (Figure 1). The higher resistance of the samples containing BT and TiO_2_ (P2, P1-TiO_2_, and P2-TiO_2_) was confirmed by compression tests, indicating that, in this case, the reinforcing effect of the two nanofillers (BT and TiO_2_) fused to ensure superior mechanical performance.

The frequency–sweep test offers information regarding the viscoelastic properties of the nanocomposite hydrogels. The storage modulus was higher than the loss modulus for all hydrogel samples, indicating their “solid-like” [72,73] structure but also a high elastic behavior [54]. The relatively high crosslinked character of the nanocomposite hydrogels is indicated by the linearity of the G’ plots (frequency independent). Still, at higher frequencies, G’ displayed a decreasing tendency, which was more visible for samples P1, P2-ZnO, and P2, probably because the hydrogel begins its transition from an elastic state towards a plastic flow state. Storage and loss modulus plots did not overlap in the measured frequency range [72]; therefore, these materials possessed an entangled elastic network. The G’ values were approximately one order of magnitude higher than G” values, comparable to the characteristics reported for naturally occurring tissues [74]. Even if all samples exhibited similar viscoelastic patterns, samples P2-TiO_2_, P1-TiO_2_, P2-ZnO, and P2 led to higher G’ and G” values than P1 and P1-ZnO, indicating a stiffening effect and higher mechanical strength with the reinforcing contribution of the nanofillers, which also provide localized regions of enhanced strength [75]. The rate at which the enclosed dynamic bonds or entanglements are broken/recombined/repositioned determines how effectively these hydrogels dissipate energy, consequently, influencing the resulting G” values. It was observed that the G” values slightly increased at frequencies lower than 1 Hz, because the samples had a longer time to flow. In contrast, values of G” started to decrease at higher frequencies, because the nanocomposites became less efficient in dissipating the energy due to the shorter times available for the polymeric network’s relaxation. The frequency sweep rheograms aligned with the stress-strain (tensile and compressive) plots.

Based on the results obtained, samples P1, P2, P2-TiO_2_, and P2-ZnO were further selected to investigate their potential as drug carriers.

In the last part of this study, the potential of these nanocomposite hydrogels to be used in biomedical applications was evaluated through UV-Vis monitoring of drug loading/release experiments and microbiological assays.

### 3.3. Nafcillin Loading and the Evaluation of Drug Release Kinetics

The nafcillin loading efficiency, presented in Appendix A, was situated between 18.7 and 24.8%. This efficiency was determined using the large volume of the nafcillin molecule, which makes its diffusion inside the crosslinked polymer matrix slightly more difficult [76,77]. However, the content absorbed ensured a good antimicrobial activity of the material during the release stage.

The results obtained via UV-Vis for estimating the release kinetics of the active ingredient (nafcillin, loaded in the nanocomposite hydrogels) are detailed below. Given that the presence of bacteria causes infected surface wounds to move toward an alkaline pH (above the typical skin pH range of 4.0–6.5), the release experiments were carried out in an alkaline medium (PBS at 7.4 pH) at a temperature of 37 °C to simulate the minimal wound environment [16]. The data analysis presented in Figure 6, Table 2, and Appendix A reveals that higher R^2^ values were obtained for the Korsmeyer–Peppas release model [78]. The value of the n parameter was situated between 0.5 and 1, which is specific to a non-Fickian, analogous drug release mechanism [79,80,81,82]. The P1 (*n* = 0.69) and P2 (*n* = 0.73) samples displayed a comparable release pattern. The samples containing bentonite seemed to exhibit lower release rates, probably due to the supplementary interactions established between the nanoclay and the antibiotic. Yet, P2-TiO_2_ exhibited a considerably higher release rate in the first 2 h, which could be very useful for potential application as a wound dressing material because wounds typically need a higher antibiotic dosage at the beginning of the treatment [83]. The larger pores of P2-TiO_2_ (Figure 3D and Appendix A), the smoother surface, and the larger dimensions of the TiO_2_ nanoparticles (Figure 2) might explain the distinct diffusion paths and faster release of the drug molecule in this case. Furthermore, according to TEM images (Figure 2), the ZnO nanoparticles possessed an irregular surface and had relatively smaller dimensions than the TiO_2_ nanoparticles, so they ensured a larger specific surface area for the adsorption of the antibiotic, which may have led to prolonged interactions with nafcillin and slower release rates for P2-ZnO. In conclusion, the variable porosity of the hydrogel, the dissimilar adsorptive performances of the nanofillers, and the specific interactions established by the antibiotic with each nanocomposite component will influence the availability of the drug by differently directing its diffusion through the polymer network [43]. The Korsmeyer–Peppas model was found to best match the nafcillin release profile based on the results of the UV-VIS monitoring.

### 3.4. Evaluation of the Antimicrobial Activity of the Nanocomposite Hydrogels

The final step in this research was to assess the antimicrobial effectiveness of nafcillin-loaded nanocomposite hydrogels. Table 3 summarizes the samples utilized for the microbiological assays. The results obtained through the “agar disc diffusion method” and “time-kill” test are further detailed.

(A)Agar disc diffusion method

For evaluating the antimicrobial activity of the nafcillin-loaded hydrogel nanocomposites through the agar disc diffusion method, the samples were placed on representative Gram-positive and Gram-negative strains, as detailed in Section 2. All four tested nafcillin-loaded samples revealed a strong antimicrobial activity against *S. aureus*, comparable to the reference antibiotic used as the positive control (chloramphenicol), as visible in Figure 7A,B. This remarkable antimicrobial activity can be explained by the synergistic action of nafcillin and the nanofillers. Typically, an antimicrobial agent diffuses into the agar and inhibits the germination and growth of the tested microorganism. Therefore, the inhibition growth zones offer evidence of the efficacy of the antimicrobial agent. In our case, the diameters of the inhibition growth zones seemed to exceed the diameter of the CP sample. However, since the diameters of the inhibition growth zones exceeded ~29 mm [84,85,86,87] and the inhibition growth zones even overlapped, we can affirm that the tested nafcillin-loaded nanocomposite hydrogels proved that they possess outstanding antibacterial activity against *S. aureus*, which recommends them for further biomedical applications (e.g., wound dressings and biological decontamination applications).

In contrast, when testing the nanocomposite hydrogels on *E. coli*, they displayed a very thin ring of inhibition, which suggests a very low susceptibility of this Gram-negative microorganism. Nonetheless, we expected these results for *E. coli*, since nafcillin is a narrow-spectrum [11] antibiotic designed for staphylococcal infections. Therefore, no inhibition can be reported for the samples tested against *E. coli*, most likely due to the specific spectrum of nafcillin.

Because this technique cannot distinguish between bactericidal and bacteriostatic effects, supplementary investigations were necessary, so we further carried out a time-kill assay.

(B)Time-kill assay

A time-kill assay was performed investigating two different contact times (1 h and 24 h) between the representative strains and the nanocomposite hydrogels, and the bacterial growth was evaluated. The results obtained for the time-kill test are summarized in Table 4 and graphically compared in Figure 8. According to these results, all four samples demonstrated good antimicrobial effects against *S. aureus* but no activity on *E. coli*. The nanocomposite hydrogels P1 (sample I) and P2-TiO_2_ (sample II.1) exhibited the highest antimicrobial activity (decrease in growth between 3log_10_ and 2log_10_) after direct contact with *S. aureus* (Table 4). For P2 (sample II) and P2-ZnO (sample II.2), the antimicrobial activity against *S. aureus* was slightly lower. These microbiological findings align with the results obtained for the nafcillin release profile.

The inhibition of the growth of Gram-positive bacteria may be caused by a specific and combined interaction between nafcillin and TiO_2_/ZnO nanoparticles with the tested microorganisms [88]. As expected, the time-kill test revealed no antimicrobial action against *E. coli*, given that nafcillin proved its efficacy specifically as an anti-staphylococcal antibiotic (Gram-positive cocci and staphylococci that produce penicillinase) [45]. However, the presence of a very weak inhibition ring of Gram-negative bacteria growth could be explained through some feeble interactions that may occur with the active ingredients, yet insufficient to stop the growth of *E. coli*. Our results are supported by the literature, which states that nafcillin is active against Gram-positive species (e.g., staphylococci, streptococci, and pneumococci), but like other penicillinase-resistant penicillins, it shows little or no activity against Gram-negative microbes [89].

The microbiological assay also showed that after 24 h, the surviving microorganisms began to divide and multiply, probably because after the elution of nafcillin by numerous cells, the high concentration of NPs/antibiotic cannot be maintained for a longer time, so the phenomenon of the increasing number of microorganisms may appear [26] after several hours. However, if such hydrogel films were applied to a wound, the samples would be considerably larger, thus retaining and releasing a higher quantity of antibiotics. For an antibiotic to work effectively, the antibiotic should remain at the binding site for a sufficient period of time in order for the metabolic processes of the bacteria to be sufficiently inhibited [90]. Beta-lactam antibiotics show time-dependent killing and produce prolonged post-antibiotic effects only with staphylococci. The frequency of drug administration is an important determinant of outcome for these drugs, as the duration of time serum levels exceed the MIC is the major determinant of efficacy [91]. Consequently, the objectives of future studies include refining the microbiological assays to identify the ideal interval of time during which the drug-loaded nanocomposite hydrogels are still effective.

## 4. Conclusions

In this study, nafcillin-loaded nanocomposite hydrogels were obtained via free-radical photopolymerization.

TEM and SEM imaging offered information on the shape and dimensions of the nanofillers and allowed the evaluation of the distinct morphology of the nanocomposite materials. The swelling capacity investigation showed that the increase in nanofiller concentration led to a lower water uptake capacity. The UV-Vis monitoring of the nafcillin release showed that the Korsmeyer–Peppas model matched the profile better.

The antibacterial potential was evaluated, and the nanocomposite hydrogels loaded with nafcillin were found to provide a strong antimicrobial activity against Gram-positive cocci but no antimicrobial activity against Gram-negative bacteria. The results obtained by microbiology-applied methods align with data found in the literature for nafcillin and TiO_2_ or ZnO nanoparticles.

## Data Availability

Data is contained within the article or Appendix A.

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
