# Peer review of "Nafcillin-Loaded Photocrosslinkable Nanocomposite Hydrogels for Biomedical Applications"

_pharmaceutics, 2023, doi:10.3390/pharmaceutics15061588_

Round 1

Reviewer 1 Report

This is an interesting work about preparation and characterization of Nafcillin-loaded photocrosslinkable nanocomposite hydrogels for anti-bacteria applications. The synthetic methodology is clearly described and the application of the materials is demonstrated. However, I still have some minor concerns before I can recommend it to be published.

(1) In the manuscript, it does not show the reproducibility of the materials. For example, in Figure 6. Could the authors show error bar in the releasing profile?

(2) To further confirm the TiO2/polymer formation, other than the TEM image, the authors could do XPS to further confirm it.

(3) The loading of antibiotics to hydrogel for wound heeling applications has been widely explored in previous literature. What's the advantage of this materials? I think the author should emphasize the significance of this work better.

Based on the above concerns, I suggest a minor revision.

Author Response

Response to reviewers’ comments

We would like to begin this reply by thanking the reviewers for taking the time to assess the manuscript and for the useful remarks. Below, marked in green, are the responses to each observation/suggestion made by the reviewers. The added or modified information was highlighted in the manuscript – marked for review only, for ease of localization.

This is an interesting work about preparation and characterization of Nafcillin-loaded photocrosslinkable nanocomposite hydrogels for anti-bacteria applications. The synthetic methodology is clearly described and the application of the materials is demonstrated. However, I still have some minor concerns before I can recommend it to be published.

  • In the manuscript, it does not show the reproducibility of the materials. For example, in Figure 6. Could the authors show error bar in the releasing profile?

Figure 6 was modified to display the error bars in accordance with the reviewer’s suggestion.

  • To further confirm the TiO2/polymer formation, other than the TEM image, the authors could do XPS to further confirm it.

We performed XRD and FT-IR analysis on the polymer composites, however the characteristic signals for TiO2 or ZnO were not observed which is due to the particles being wrapped by the polymer.

  • The loading of antibiotics to hydrogel for wound heeling applications has been widely explored in previous literature. What's the advantage of this materials? I think the author should emphasize the significance of this work better.

The introduction section was improved as suggested by the reviewer.

Based on the above concerns, I suggest a minor revision

Reviewer 2 Report

Authors fabricated Nafcillin-Loaded Photocrosslinkable hydrogel. This study is interesting however authors failed to study its primary characteristics. Therefore, this manuscript is not considered at this stage. Improve the manuscript as per my following comments.

 The abstract has limited information; need to rewrite it. Generally abstract contains background sentences, methods and significant findings.

At present, the introduction is too much; the authors just collected relevant content related to this study rather than highlighting the exact problem and novelty. So please rewrite it preciously.

Line 166/167, mention the specific goal of this paper with the name of the nanocomposite.

Line 172, make space between the number and oC. Check other places in the manuscript.

How authors know the successful free radical photopolymerization happens with interaction results such as FTIR or NMR data. SEM doesn’t provide this details information. Without these data, this manuscript is not worth publishing.

No drug-loading data in the manuscript.

Mention UV wavelength for photopolymerization. Details need for the readers.

Cite the suggested manuscript in the text; Journal of the Taiwan Institute of Chemical Engineers, 134, 104301 (2022), Biotechnology and Bioengineering 118 (12), (2021) 4590-4622, Soft Matter 16 (6) (2020) 1404-1454 Polymers, 15(1), 132 (2023)

Provide the swelling calculation formula in the manuscript and describe it briefly.

Why do authors highlight the abbreviation of the chemical name? No need.

In this TiO2 and ZnO, wt% was taken based on which raw materials. Mention it in the manuscript.

In Wt.%, remove dot mark.

Ideally the conclusion should one or two paragraphs with limited text. At present, you conclusion is too lengthy. Please rewrite it preciously.

In the antibacterial assay, please provide an exact mechanism.

All instruments’ details need to mention in the manuscript. Some instrument information is missing.

Avoid citations in the conclusion part, line 567.

English needs to be improved moderately. 

Author Response

Response to reviewers’ comments

We would like to begin this reply by thanking the reviewers for taking the time to assess the manuscript and for the useful remarks. Below, marked in green, are the responses to each observation/suggestion made by the reviewers. The added or modified information was highlighted in the manuscript – marked for review only, for ease of localization.

Authors fabricated Nafcillin-Loaded Photocrosslinkable hydrogel. This study is interesting however authors failed to study its primary characteristics. Therefore, this manuscript is not considered at this stage. Improve the manuscript as per my following comments.

 The abstract has limited information; need to rewrite it. Generally abstract contains background sentences, methods and significant findings.

The abstract was re-written in accordance with the reviewer’s suggestion.

At present, the introduction is too much; the authors just collected relevant content related to this study rather than highlighting the exact problem and novelty. So please rewrite it preciously.

Additional references were added in the introduction section as suggested by the reviewer.

Line 166/167, mention the specific goal of this paper with the name of the nanocomposite.

The novelty section was improved in accordance with the reviewer’s suggestion.

Line 172, make space between the number and oC. Check other places in the manuscript.

We have performed a manuscript spelling and style re-check in accordance with the reviewer’s suggestion.

How authors know the successful free radical photopolymerization happens with interaction results such as FTIR or NMR data. SEM doesn’t provide this details information. Without these data, this manuscript is not worth publishing.

By photopolymerization a cross-linked polymeric material is obtained. NMR characterization is difficult due to the lack of solubility. The FT-IR analysis was performed, however, due to the low concentration of filler agents and their wrapping by the polymer only the specific signal of the polymer components can be observed. Please see the FT-IR below, it confirms the formation of the polymeric structure, but the presence of the filler cannot be properly ascertained.

No drug-loading data in the manuscript.

The loading kinetics were not studied, however, the details regarding the concentration used for loading are presented in the manuscript, please see section “2.2.2.     Nafcillin loading/release tests”. The goal was to observe the release of the drug and its efficacy.

Mention UV wavelength for photopolymerization. Details need for the readers.

The information was present in the manuscript “low-pressure Hg UV lamp, λem = 254 nm”.

Cite the suggested manuscript in the text; Journal of the Taiwan Institute of Chemical Engineers, 134, 104301 (2022), Biotechnology and Bioengineering 118 (12), (2021) 4590-4622, Soft Matter 16 (6) (2020) 1404-1454 Polymers, 15(1), 132 (2023)

The reference list was improved as suggested by the reviewer.

Provide the swelling calculation formula in the manuscript and describe it briefly.

The formula was added in accordance with the reviewer’s suggestion.

Why do authors highlight the abbreviation of the chemical name? No need.

We have removed the highlight.

In this TiOand ZnO, wt% was taken based on which raw materials. Mention it in the manuscript.

The TiO2 and ZnO content was calculated based on the total amount in the reaction and it was already marked in Table 1 (from the total mass of the reaction mixture)

In Wt.%, remove dot mark.

We have made the modification as suggested by the reviewer.

Ideally the conclusion should one or two paragraphs with limited text. At present, you conclusion is too lengthy. Please rewrite it preciously.

We have tried to improve the presentation of the conclusion.

In the antibacterial assay, please provide an exact mechanism.

Additional information was added about the antimicrobial assay.

All instruments’ details need to mention in the manuscript. Some instrument information is missing.

We have rechecked the methods section and improved the presentation.

Avoid citations in the conclusion part, line 567.

The conclusion section was improved.

Round 2

Reviewer 2 Report

The authors improved the manuscript; however many important concerns are not complied.

1.How authors know the successful free radical photopolymerization happens with interaction results such as FTIR or NMR data. SEM doesn’t provide this details information. Without these data, this manuscript is not worth publishing.

Authors need to incorporate the right data in the manuscript for the reader’s understanding. Without this, it is impossible to recommend publishing this manuscript in this prestigious journal (Q1; IF 6.5). Your FTIR data doesn’t provide anything for the material preparation.

2. Conclusion section should be one paragraph. At present so many texts. Make it short with the important findings of this study.

3. In the introduction there are many paragraphs merged into 4 or 5 paragraphs.

4. Introduction is section 1. Materials and methods should be 2 and so on…. Check it carefully.

5. Authors must provide drug loading, Encapsulation efficiency and release data in %. Without knowing the loading efficiency, how do authors move to another experiment?

English needs to be checked. Bold wording needs to remove. 

Author Response

Response to reviewers’ comments

We would like to begin this reply by thanking the reviewer for taking the time to reassess the manuscript and for the useful remarks. Below, marked in green, are the responses to each observation/suggestion made by the reviewers. The added or modified information was highlighted in the manuscript – marked for review only, for ease of localization.

The authors improved the manuscript; however many important concerns are not complied.

1.How authors know the successful free radical photopolymerization happens with interaction results such as FTIR or NMR data. SEM doesn’t provide this details information. Without these data, this manuscript is not worth publishing.

We have added the FTIR analysis spectra of the xerogels in the supporting info file (Figure S1). It presents the lack of vinyl bonds and the characteristic signals for pyrrolidone units. NMR is not possible for us due to the lack of solubility of the crosslinked xerogels. We do not have access to solid state NMR equipment. A short discussion was added in the manuscript as suggested.

Authors need to incorporate the right data in the manuscript for the reader’s understanding. Without this, it is impossible to recommend publishing this manuscript in this prestigious journal (Q1; IF 6.5). Your FTIR data doesn’t provide anything for the material preparation.

  1. Conclusion section should be one paragraph. At present so many texts. Make it short with the important findings of this study.

We have completely rewritten the conclusion section in accordance with the reviewer’s suggestion.

  1. In the introduction there are many paragraphs merged into 4 or 5 paragraphs.

We have condensed the introduction section into 5 paragraphs as suggested.

  1. Introduction is section 1. Materials and methods should be 2 and so on…. Check it carefully.

There was an error when the pdf was generated from our word file. We have corrected the issue.

  1. Authors must provide drug loading, Encapsulation efficiency and release data in %. Without knowing the loading efficiency, how do authors move to another experiment?

The loading efficiency was added in the supporting file as Figure S10 and a short discussion was added in the manuscript.

Round 3

Reviewer 2 Report

The authors improved the manuscript to a satisfactory level.

Check minor errors.